# Sexual Dimorphism in Extracellular Matrix Composition and Viscoelasticity of the Healthy and Inflamed Mouse Brain

**DOI:** 10.3390/biology11020230

**Published:** 2022-01-31

**Authors:** Clara Sophie Batzdorf, Anna Sophie Morr, Gergely Bertalan, Ingolf Sack, Rafaela Vieira Silva, Carmen Infante-Duarte

**Affiliations:** 1Experimental and Clinical Research Center, Max Delbrück Center for Molecular Medicine and Charité—Universitätsmedizin Berlin, Corporate Member of Freie Universität Berlin and Humboldt-Universität zu Berlin, Lindenberger Weg 80, 13125 Berlin, Germany; clara.batzdorf@charite.de (C.S.B.); rafaela.vieira-da-silva@charite.de (R.V.S.); 2Department of Radiology, Charité—Universitätsmedizin Berlin, Corporate Member of Freie Universität Berlin and Humboldt-Universität zu Berlin, Charitéplatz 1, 10117 Berlin, Germany; anna-sophie.morr@charite.de (A.S.M.); gergely.bertalan@charite.de (G.B.); ingolf.sack@charite.de (I.S.); 3Einstein Center for Neurosciences Berlin, Charité—Universitätsmedizin Berlin, Corporate Member of Freie Universität Berlin and Humboldt-Universität zu Berlin, Charitéplatz 1, 10117 Berlin, Germany

**Keywords:** multiple sclerosis, experimental autoimmune encephalomyelitis, sexual dimorphism, brain viscoelasticity, magnetic resonance elastography, extracellular matrix, cerebral cortex, neuroinflammation, basement membrane

## Abstract

**Simple Summary:**

In multiple sclerosis (MS), an autoimmune disease of the central nervous system that primarily affects women, gender differences in disease course and in brain softening have been reported. It has been shown that the molecular network found between the cells of the tissue, the extracellular matrix (ECM), influences tissue stiffness. However, it is still unclear if sex influences ECM composition. Therefore, here we investigated how brain ECM and stiffness differ between sexes in the healthy mouse, and in an MS mouse model. We applied multifrequency magnetic resonance elastography and gene expression analysis for associating in vivo brain stiffness with ECM protein content in the brain, such as collagen and laminin. We found that the cortex was softer in males than in females in both healthy and sick mice. Softening was associated with sex differences in expression levels of collagen and laminin. Our findings underscore the importance of considering sex when studying the constitution of brain tissue in health and disease, particularly when investigating the processes underlying gender differences in MS.

**Abstract:**

Magnetic resonance elastography (MRE) has revealed sexual dimorphism in brain stiffness in healthy individuals and multiple sclerosis (MS) patients. In an animal model of MS, named experimental autoimmune encephalomyelitis (EAE), we have previously shown that inflammation-induced brain softening was associated with alterations of the extracellular matrix (ECM). However, it remained unclear whether the brain ECM presents sex-specific properties that can be visualized by MRE. Therefore, here we aimed at quantifying sexual dimorphism in brain viscoelasticity in association with ECM changes in healthy and inflamed brains. Multifrequency MRE was applied to the midbrain of healthy and EAE mice of both sexes to quantitatively map regional stiffness. To define differences in brain ECM composition, the gene expression of the key basement membrane components laminin (*Lama4, Lama5*), collagen (*Col4a1, Col1a1*), and fibronectin (*Fn1*) were investigated by RT-qPCR. We showed that the healthy male cortex expressed less *Lama4*, *Lama5*, and *Col4a1*, but more *Fn1* (all *p* < 0.05) than the healthy female cortex, which was associated with 9% softer properties (*p* = 0.044) in that region. At peak EAE cortical softening was similar in both sexes compared to healthy tissue, with an 8% difference remaining between males and females (*p* = 0.006). Cortical *Lama4*, *Lama5* and *Col4a1* expression increased 2 to 3-fold in EAE in both sexes while *Fn1* decreased only in males (all *p* < 0.05). No significant sex differences in stiffness were detected in other brain regions. In conclusion, sexual dimorphism in the ECM composition of cortical tissue in the mouse brain is reflected by in vivo stiffness measured with MRE and should be considered in future studies by sex-specific reference values.

## 1. Introduction

Multiple Sclerosis (MS) is a chronic inflammatory demyelinating disease of the central nervous system (CNS) affecting approximately 2.8 million people worldwide [1]. It displays a prominent sexual dimorphism in relation to susceptibility, incidence, pathology, and progression [2]. In women, the risk of MS is three times higher than in men, with disease onset occurring at earlier ages, but, generally, with slower progression [3,4,5,6,7,8]. While the higher susceptibility in women is associated with sex hormones and a differential immune response to inflammation, worsened progression in men is linked to enhanced neurodegeneration [2,7,9]. Moreover, MS incidence in women has been increasing over the last decades probably due to a sex-dependent response to environmental and lifestyle factors [5,6], with a current estimated female to male ratio of 2–4:1 [1,10]. 

The mechanical properties of the brain investigated by magnetic resonance elastography (MRE) also display sex differences. MRE is a non-invasive method that, based on the induction of shear waves through harmonic vibrations, allows the assessment of viscoelastic properties of the brain tissue in vivo [11]. MRE of healthy individuals showed that male brains were softer than female brains [12,13,14]. In response to inflammation, however, the brains of female MS patients exhibited stronger reduction in viscoelasticity compared to age-matched healthy males [12,15,16]. Similarly, in the MS mouse model, experimental autoimmune encephalomyelitis (EAE), we previously demonstrated a reduction in brain stiffness in sick mice [17,18,19]. This softening behavior seemed to be associated with inflammatory processes such as immune cell infiltration, demyelination, and loss of the blood–brain barrier (BBB) integrity, as well as remodeling of the extracellular matrix (ECM) [18,19,20,21]. We also recently showed that brain viscoelastic properties seem to be particularly dependent on ECM organization and axonal structure [22].

Furthermore, in MS, the ECM composition at lesion sites appears to be altered. Active lesions are marked, among others, by the deposition of fibronectin, correlating with BBB disruption and the accumulation of laminin and collagen type IV in the basement membrane and perivascular cuffs [18,23,24]. Moreover, it has been reported that estrogen can influence the production of matrix metalloproteinases [25,26], indicating that the degradation of ECM components may be influenced by sex. 

Laminins, collagens, and fibronectin are mainly found in the ECM of the basement membrane in the brain and play an important role in the regulation of the BBB integrity [27]. Furthermore, collagen type I provides tensile strength and stiffness, while collagen type IV is responsible for flexible networks [28]. However, how sex-specific differences influence neuroinflammatory remodeling of the ECM and, consequently, viscoelastic brain properties, remains unclear.

Therefore, the aim of this study was to investigate sex effects on brain viscoelasticity in healthy and EAE mice via multifrequency MRE, which allows the accurate analysis of small cerebral regions [29], and to analyze sexual dimorphism in the remodeling of brain ECM.

## 2. Materials and Methods

### 2.1. Animals and EAE Model

All animal experiments were approved by the Berlin State Office for Health and Social Affairs (LAGeSo, G106/19) and conducted in strict adherence to the European guidelines for the care and use of laboratory animals under directive 2010/63/EU of the European Parliament and of the Council of 22 September 2010.

SJL mice that were 10–15 weeks old (Janvier, SAS, Le Genest Saint Isle, France), and are known for exhibiting sex-dependent susceptibility to EAE induction [30,31], were investigated. The mice were housed under standard conditions with a 12:12h light–dark-cycle and ad libitum access to food and water. The four experimental groups consisted of female healthy (*n* = 14), male healthy (*n* = 14), female EAE (*n* = 23) and male EAE (*n* = 19) mice. To induce EAE, the animals were immunized with 250 μg of proteolipid peptide PLP_139–151_, emulsified in 100 μL complete Freund’s adjuvant (Thermo Fisher Scientific, Waltham, MA, USA) and 800 μg Mycobacterium tuberculosis H37Ra (Difco, Detroit, MI, USA). Furthermore, 250 ng of pertussis toxin (List Biological Laboratories, Campbell, CA, USA) resuspended in phosphate-buffered saline (PBS) (Gibco, Grand Island, NY, USA) was injected intraperitoneally on day 0 and day 2 after immunization. Mice were monitored daily for clinical signs and scored as follows: 0.5—tail paresis or weak righting reflex; 1—tail plegia or tail paresis and weak righting reflex; 1.5—tail plegia and weak righting reflex; 2.0—additional hind limb paresis; 3.0—paraplegia; 4.0—additional forelimb paresis; 5.0—moribund or dead animal. To comply with animal welfare guidelines, all mice with a score higher than 3 were euthanized and removed from the study (*n*_female_ = 2, *n*_male_ = 1). The EAE model using SJL mice has been shown to be more suitable for cerebral MRE studies than EAE in C57BL/6, since PLP-induced EAE in SJL consistently leads to the development of brain lesions and to elasticity alterations in brain structures [17,18,19,32,33].

### 2.2. MRE Acquisition

MRE acquisition was performed on a preclinical 7 Tesla MRI scanner (BioSpec, Bruker, Ettlingen, Germany) operated with ParaVision 6.1 software (Bruker, Billerica, MA, USA) using a 20-mm diameter 1H-RF quadrature volume coil (RAPID Biomedical, Rimpar, Germany). The set up was similar to that previously described [19,29]. EAE mice were scanned based on their individual score at the peak of disease around day 10–12 after immunization. Age-matched healthy control mice were scanned on the same experimental days. To this end, mice were placed on a custom-built animal holder and anesthetized with 1.5–2.0% isoflurane in 30% O_2_ and 70% N_2_O by mask under continuous respiratory monitoring, using a pressure-sensitive pad placed on the dorsal thorax (Small Animal Instruments Inc., Stony Brook, NY, USA). Body temperature was monitored using a rectal probe and kept constant by circulating warm water through pads integrated in the animal holder. External vibrations were created by a custom-made driver system, using a non-magnetic piezoceramic actuator, and translated to the skull via a transducer rod to the head cradle of the mouse to induce shear waves in the brain [19]. Multifrequency single-shot MRE was conducted by consecutively exciting five external vibrational frequencies (1000, 1100, 1200, 1300, and 1400 Hz) which were encoded by a single-shot EPI sequence. Seven coronal slices with a slice thickness of 0.8 mm and a 0.18 mm × 0.18 mm spatial resolution were acquired. The covered brain region was consistent with previous MRE studies [19,29], which facilitated the comparison of values. Further imaging parameters were TA = 9 min, TE = 53 ms, TR = 4000 ms, FOV = 16.2 mm × 10.8 mm, and matrix size = 90 × 60. 

### 2.3. Data Reconstruction

Multifrequency MRE data were reconstructed using the tomoelastography post-processing pipeline, as described previously [29]. Viscoelasticity parameter maps were obtained based on shear wave speed *c* (in m/s), as a marker for tissue stiffness, and phase angle *φ* (in rad, also denoted as loss angle of the complex shear modulus), as a marker of tissue fluidity. Regions of interest (cerebral cortex, hippocampus, thalamic area, whole coronal midbrain slice) were defined manually, according to anatomical structures in a blinded fashion using MATLAB (Version 9.4 (R2018a). The MathWorks Inc.; 2018, Natick, MA, USA). Due to technical errors, *n*_male naïve_ = 1 had to be excluded from the MRE analysis.

### 2.4. Tissue Processing

Animals were sacrificed directly after the MRE measurements with an overdose of ketamine (Inresa Arzneimittel GmbH, Freiburg im Breisgau, Germany) and xylazine (CP-Pharma, Burgdorf, Germany) followed by cardiac perfusion with PBS (Gibco, Grand Island, NY, USA). The brains were extracted and one hemisphere was preserved in 4% paraformaldehyde (PFA) (Carl Roth, Karlsruhe, Germany) overnight at 4 °C, following dehydration in 30% sucrose at 4 °C. Afterwards, the tissue was embedded in O.C.T. (Sakura Finetek, Tokyo, Japan) and stored at −80 °C until preparation for histological staining. From the other half of the brain, the cerebral cortex and hippocampus were carefully extracted under a microscope, then freeze-dried in liquid nitrogen and stored at −80 °C for gene expression analysis. Post-MRE tissue processing was performed on a subgroup of animals (*n*_female naïve_ = 6, *n*_male naïve_ = 6, *n*_female EAE_ = 6, *n*_male EAE_ = 7).

### 2.5. Gene Expression Analysis

RNA was extracted from the hippocampus and the cerebral cortex using the Quick-RNA-MiniPrep Kit (Zymo Research, Irvine, CA, USA), and cDNA synthesis was carried out with the High-Capacity cDNA Reverse Transcription Kit (Thermo Fisher Scientific, USA), according to the manufacturer’s instructions. Quantitative RT-PCR was performed with the QuantStudio 6 Flex Real-Time PCR system (Thermo Fisher Scientific, USA) using TaqMan® probes for the following genes: *Lama4, Lama5, Col4a1, Col1a1,* and *Fn1* (Thermo Fisher Scientific, USA; Appendix A)*. Hprt1* served as the endogenous reference [34,35]. Collagens (*Col4a1*, *Col1a1*) provide fibrillar and net-like structures [28], while laminins (*Lama4*, *Lama5*) and fibronectin (*Fn1*), which are glycoproteins, are important for cellular attachment to the matrix [36,37] and influence collagen organization [38]. Therefore, we considered these proteins to be important regulators of the mechanical properties of the brain ECM. 

### 2.6. Histology

For immunofluorescence staining the tissue was cut into 6 µm slices, thawed, fixed with 4% PFA for 15 min and then blocked with PBS containing 8% horse serum (Gibco, USA), 3% bovine serum albumin (Sigma-Aldrich, Darmstadt, Germany) and 1% Triton TM X-100 (Thermo Fisher Scientific, USA) for one hour at room temperature. Sections were then incubated overnight with the primary antibodies at 4 °C, diluted in PBS containing 20% blocking solution. As primary antibodies, we used mouse anti-fibronectin, 1:400 (Novus Biologicals, 2755-8, Littleton, CO, USA), rabbit monoclonal anti-collagen IV (EPR22911-127), 1:400 (Abcam, ab236640, Cambridge, UK) and rat monoclonal anti-CD3 (17A2), 1:100 (Invitrogen, 14-0032-82, Carlsbad, CA, USA). On the following day, sections were incubated with secondary antibodies: anti-rat, anti-mouse, and anti-rabbit, conjugated with Alexa-Fluor 488 or 568, 1:500 (Invitrogen, Carlsbad, CA, USA) for one hour at room temperature and counterstained with 4′,6-diamidino-2-phenylindole (DAPI) at 1:10,000. Images were acquired at 20 times magnification with the Keyence Fluorescence Microscope BZ-X800 (Keyence Corporation, Osaka, Japan).

### 2.7. Statistical Analysis

Analyses were performed using GraphPad Prism 9.3 (GraphPad Software, La Jolla, CA, USA) with significance levels defined as * *p* < 0.05, ** *p* < 0.01, *** *p* < 0.001. All *p*-values below 0.1 are reported. Statistical group comparison was performed with an unpaired two-tailed *t* test, or with two-way analysis of variance (ANOVA) with post-hoc pairwise comparisons where appropriate. Values are reported with means, standard deviation (SD) and a 95% confidence interval (CI). Graphs are reported with means and a 95% CI.

## 3. Results

### 3.1. Sex-Specific Viscoelastic Properties of Healthy Mouse Brain

To determine sex differences in the brain’s mechanical properties, the viscoelasticity of healthy female (
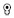
) and male (
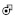
) midbrains were compared. The predefined regions of interest were averaged for mean values of shear wave speed *c* (in m/s) and fluidity *φ* (in rad). Representative MRE parameter maps for the whole midbrain slice covering the cortex, hippocampus, and thalamus are depicted in Figure 1.

No differences in *c* or *φ* were observed when comparing the whole midbrain area of males and females (Figure 2A,B). However, sex became an important variable when considering the cortical viscoelasticity, which was on average 9% softer in males than in healthy age-matched females (mean *c*_female_ = 2.90 ± 0.33 m/s, 95%CI 2.71–3.09 m/s; mean *c*_male_ = 2.65 ± 0.29 m/s, 95%CI 2.63–2.78 m/s; *p* = 0.044). No sex-related differences in *c* or *φ* for the hippocampus or the thalamic area were detected.

### 3.2. Differences in Extracellular Matrix Composition in the Healthy Brain of Males and Females

Gene expressions of laminins (*Lama5* and *Lama4*), collagens (*Col4a1* and *Col1a1*), and fibronectin (*Fn1*) were quantified in the cerebral cortex and hippocampus as a control region in healthy male (*n* = 6) and age-matched female (*n* = 6) mice. Figure 3 represents relative gene expression levels in healthy males compared to females. Laminin and collagen type IV expression was significantly lower in the male cortex than in the female one (*Lama5* −4.0-fold, *p* = 0.019; *Lama4* −3.4-fold, *p* = 0.006; *Col4a1* −1.6-fold, *p* = 0.029), while collagen type I expression did not present sexual dimorphism (Figure 3A). In contrast, a higher expression of fibronectin was found in the cortex of healthy males (*Fn1* 1.9-fold, *p* = 0.002). No differences related to sex were observed in the hippocampus (Figure 3B).

### 3.3. Sex-Specific Changes of Brain Viscoelastic Properties during EAE

Next, we investigated whether viscoelasticity changes of the brain following inflammation revealed a sexual dimorphism. For that, data from healthy mice were compared with a cohort of female and male mice at the peak EAE (*n*_female_ = 21, mean score = 2.3; *n*_male_ = 18, mean score = 2.3). 

Only the male EAE group showed a 5% global reduction of stiffness in the midbrain (*c*_male naïve_ = 3.02 ± 0.19 m/s, 95%CI 2.90–3.13 m/s; *c*_male EAE_ = 2.86 ± 0.15 m/s, 95%CI 2.78–2.93 m/s; *p* = 0.024) when averaging the whole coronal section, while fluidity remained unaltered (Figure 4A). In the hippocampus, no significant stiffness changes were observed at peak EAE in both sexes, although a trend could be seen for male EAE. 

When considering different brain regions, the elasticity of the cortex was the most affected during EAE. We detected a 7% reduction for female and 6% for male mice when compared with age-matched control mice (*c*_female naïve_ = 2.90 ± 0.33 m/s, 95%CI 2.71–3.09 m/s; *c*_female EAE_ = 2.70 ± 0.16 m/s, 95%CI 2.63–2.78 m/s; *p* = 0.013; *c*_male naïve_ = 2.65 ± 0.29 m/s, 95%CI 2.63–2.78 m/s; *c*_male EAE_ = 2.49 ± 0.12 m/s, 95%CI 2.43–2.55 m/s; *p* = 0.067). The comparison between sexes revealed, however, no sex-related differences in the overall cortical softening during inflammation. The male EAE cortex was approximately 8% softer than the female EAE cortex (*p* = 0.006), maintaining the sex-dependent difference in stiffness observed in healthy mice (Figure 4B, center). In the thalamus, no inflammation-related changes of the mechanical properties were observed (Figure 4B, right). In both sexes, regional fluidity was not affected by EAE (Appendix A).

### 3.4. Extracellular Matrix Remodeling in Female and Male EAE Brains

To assess inflammation-induced changes of the ECM components that appeared to be differentially expressed in male and female tissues, we investigated the gene expression levels of laminins, collagen type I and IV and fibronectin in the cerebral cortex and hippocampus of healthy and EAE mice. As shown in Figure 5A and Table 1, at peak EAE, the cortical expression of *Lama4* and *Lama5* increased similarly in both sexes (all *p* < 0.05). Given the baseline differences between healthy animals, their expression remained significantly higher in the female EAE group compared to the male (*Lama5* 3.85-fold, *Lama4* 3.86-fold, all *p* < 0.001) (Table 1). However, for *Col4a1* we observed a 3.0-fold increase in the cortex in female EAE and 2.0-fold in male EAE (all *p* < 0.01) in comparison to the respective healthy group, augmenting the initial sex difference from 1.6-fold in healthy mice to 2.38-fold (*p* < 0.001) in EAE. In addition, a decreased expression of *Col1a1* (−2.1-fold, *p* < 0.001) and *Fn1* (−1.4-fold, *p* = 0.018) was observed in the male cortex.

In the hippocampus (Figure 5B), we observed mild effects of inflammation on the expression of the selected ECM-components. *Lama5* expression remained unaltered for males at peak EAE, but increased 1.83-fold in females (*p* = 0.064), leading to a significant 2.41-fold difference (*p* = 0.025) between the sexes in the disease state (Figure 5B). Other investigated genes did not show any significant changes in the hippocampus regarding disease status or sex.

To visualize the ECM remodeling in both sexes and confirm the gene expression data, we performed representative immunohistological stainings in the cerebral cortex of two of our targets, fibronectin and collagen type IV. A representative cortical image of each group is shown in Figure 6. Collagen type IV was present around vessel-like structures, resembling the formation of the endothelial basement membranes (Figure 6A). Corroborating gene expression data, an upregulation of collagen type IV was observed in EAE mice for both sexes, compared to their corresponding healthy controls, with especially pronounced expression in female EAE. Similarly, fibronectin was also present in vessel-like structures, as demonstrated in Figure 6B. Interestingly, in healthy males, fibronectin appeared widely distributed around cell bodies.

Additionally, to investigate whether ECM remodeling was associated with the presence of leukocyte infiltration, the cortical tissue was stained for CD3 to identify perivascular infiltrates in the cerebral cortex. Only one out of five male EAE mice presented a cortical perivascular lesion, and no leukocyte infiltrate was detected in females (Appendix A). This indicates that the observed changes affecting elasticity and ECM composition in EAE were not related to acute lesion formation but diffuse inflammatory events.

## 4. Discussion

In this study, we built on our previous work on ECM alterations during neuroinflammation investigated by MRE, and further focused on sexual dimorphism, which potentially affects ECM composition and brain viscoelasticity. Our data provided evidence of sexual dimorphism in the ECM of cortical tissue, namely with respect to basement membrane proteins that may directly impact tissue structure or, as a result of the local inflammatory process, indirectly contribute to the observed divergence in tissue stiffness and the response to neuroinflammation.

Using MRE, we demonstrated here, for the first time, that the murine healthy brain exhibits sex-dependent differences in mechanical properties. Stiffness differences seem to primarily affect the cerebral cortex, as the male healthy cortex was, on average, 9% softer than the female one. This data is in line with previous MRE studies in humans that indicated significant sex differences in healthy brain viscoelasticity, with male brains being 9 to 11% softer, on average [12,13]. 

Furthermore, we demonstrated a significantly higher expression of laminin and collagen type IV in the healthy female cortex, while fibronectin was about two times more expressed in the male cerebral cortex than in the female cortex. Laminin, collagen type IV, and fibronectin are predominantly located within the basement membrane of CNS tissue [23,27]. Since these components have been shown to have relevant mechanical features [28,39,40,41,42,43,44,45,46] (extensively reviewed by [47,48,49]), our results indicate that differences in the constitution of the basement membrane may have an impact on cortical stiffness in a sex-specific manner.

In this line, sexual dimorphism has been reported in non-neuronal ECM, such as in tendons or vocal folds [50,51]. Moreover, estradiol and testosterone have been reported to have a differential effect on fibronectin synthesis depending on the investigated cells, which is in agreement with our results on the high fibronectin to collagen ratio in the male cortex [52]. Likewise, collagen turnover seems to exhibit a strong sex-dependence across the lifespan, with pronounced changes in women around menopause [53], and the connection between collagen content in the skin and circulating estrogen levels is well established [54]. Nonetheless, we cannot exclude that cellular differences between the female and male cortex may have contributed to the observed sex-specific disparity in stiffness. Studies in stroke models suggest a correlation between stiffness and neuronal cell count [55], which has been shown to differ between sexes [56]. While men seem to have more cortical neurons, female cerebral cortices show considerably larger neuropil [56,57]. On the other hand, cellular composition directly influences ECM configuration regarding its components, as well as post-translational processing such as cross-linking [28] which, then again, influences mechanical properties [41] and thereby cannot be considered as an independent variable.

No sex-specific differences in viscoelastic properties of the hippocampus were observed in our study, although more microglia and astrocytes are present in the dentate gyrus of female mice [58], while the neuronal density in the same area seems to be higher in males [59]. In addition, we did not observe differences in viscoelastic properties of the thalamus, which is in line with previous data in humans that showed no sexual dimorphism in the deep gray area [14]. Nonetheless, future studies should address possible viscoelastic changes in thalamic subregions. This will be of particular interest since the lateral posterior nucleus, the mouse homologue for the human pulvinar nucleus, has been shown to be functionally connected to cortical areas including auditory, visual, and somatosensory cortices [60,61] and to present sex-dependent differences in connectivity [62].

Under neuroinflammatory conditions, we observed a mild effect in females and a significant softening of the midbrain of 5% in male EAE, although previous studies reported a global softening in female EAE brains [17,18,19] as well as for MS patients of both sexes [12]. This discrepancy could be explained by the limitation of the analyzed region, which was restricted to a single coronal slice, and did not cover the usually affected cerebellum or optic nerve [63,64]. Another factor to consider when comparing these results with our previous MRE studies [17,18,19,32] is our current focus on regional viscoelasticity values based on multifrequency inversion, which allows a higher resolution, and is therefore more sensitive to regional alterations, as studies in human subjects have shown [65,66,67].

When assessing regional mechanical properties, both female and male mice showed a similar degree of cortical softening during EAE. In both sexes, inflammation led to reduced stiffness, resulting in an 8% softer cortex in males than females in EAE, which is similar to the intersexual difference of 9% observed in healthy mice. No significant effects of inflammation on tissue elasticity were observed for the other investigated regions. Thus, multifrequency MRE revealed the murine cortex as the brain region with the largest sex-specific variation in stiffness under normal and neuroinflammatory conditions.

Interestingly, this effect was reflected by the gene expression of main proteins forming the ECM basement membrane, which presented pronounced sex differences in the cortex, and almost none in the hippocampus. In the cerebral cortex, both female and male EAE showed significantly higher expression of laminins and collagen type IV compared to healthy mice. This is in coherence with previous work on glial scars, attributing tissue softening to the enhanced expression of laminin and collagen type IV [68]. However, our results also revealed higher laminin and collagen expression in the healthy female cortex compared to the softer healthy male cortex. Nonetheless, previous work has indicated that tissue stiffness is intricately regulated by the interaction of cellular adhesion and matrix composition, providing seemingly paradoxical results when comparing healthy and pathological tissue properties [69]. Additionally, stiffness of collagen type IV has been shown to be dependent on the degree of crosslinking [41] which was not investigated here, and might explain the discrepancies between data in healthy and diseased tissue, since neuroinflammation triggers an increase in ECM degradation [70]. 

In both MS and EAE, deposition of basement membrane molecules associated with activated microglia in the vicinity of white matter lesions contribute to the disruption of the BBB integrity [23,24,71]. Our histological data support the elevated presence of collagen type IV around vessel-like structures in the cortex in EAE, while we did not observe perivascular leukocyte infiltration in that region. This is expected from peptide-induced EAE pathology, where inflammation mainly affects the spinal cord, cerebellum, and the optic nerve [63], and does not habitually lead to cortical lesions [64]. Pathological changes in the cortex are proposed to be directly related to spinal lesion formation via Wallerian degeneration [72], but are also highly dependent on meningeal inflammation with soluble factors leading to demyelination and neurodegeneration [73]. Therefore, we propose that the cortical changes observed here, which differ greatly from ECM remodeling described in white matter lesions [18,23], result from underlying meningeal and spinal inflammation rather than from the infiltration of leukocytes into the cortical parenchyma. 

Obviously, alterations of the basement membrane cannot solely explain the elasticity changes observed in the cortical tissue, but could rather reflect the inflammatory, activated state of matrix-producing cells, such as astrocytes, oligodendrocyte precursor or endothelial cells [74]. It is reasonable to posit that stiffness changes result, at least partially, from the inflammatory state of these cells. Nonetheless, another relevant aspect that cannot be discarded, is the influence of basement membrane components on the interstitial matrix, the largest component of the brain ECM [75]. Since the interstitial matrix is in part constituted by laminins and collagens [27], it is possible that enhanced expression of such components at vascular sites leads to a partial diffusion into the parenchymal space [74]. These molecules could then interact with and alter the ECM of other compartments, such as the perineuronal matrix, resulting in a major structural effect contributing to the elastic changes here discussed.

The inflammation-induced upregulation of *Lama4* and *Lama5* was about 3-fold in both sexes. However, considering baseline differences, their expression was still about 4-fold higher in female EAE than in male EAE, demonstrating that the effect size of the inherent sex differences was larger than the observed inflammatory effect. In addition, we observed a downregulation of *Col1a1* and *Fn1* in EAE male cortices. The histological correlates showed a reduction of fibronectin in the inflamed male cortex compared to the healthy group, affecting not only the vasculature but also fibronectin in association with cortical nuclei. This is in contrast with previous research, which described the deposition of fibronectin in MS lesions [23,24] in association with perivascular infiltrates and tissue softening [18]. Given the absence of leukocyte infiltrates in the cortex, it is conceivable that cortical fibronectin and collagen type I levels were not increased, and may not have contributed to cortical softening. As most of our knowledge about neuroinflammatory ECM remodeling stems from white matter lesions [23], ECM changes in normal-appearing gray matter are still not well understood.

In addition, none of the investigated regions exhibited changes in tissue fluidity *φ* between naive and EAE mice of the same sex, which has been equally described in our previous studies in acute EAE [17,18,19,32]. Tissue fluidity is influenced by the motility of structure elements in the viscoelastic network [76] and, thus, seems to reflect cellular confinement [77]. This confinement is affected by the interaction and organization of macromolecular networks in the ECM [77,78]. Since we did not observe any changes in fluidity properties between healthy and EAE brains across all regions, we expect the complexity of the tissue architecture to be similar between sexes and not affected by inflammation. 

Taken together, our data on the cortical remodeling of the ECM in the context of EAE suggests sex-specific processes. Sex-dependent changes in the expression levels of basement membrane proteins seem to have the same net effect on macroscopic viscoelastic properties of the cortex because we observed a similar amount of softening in both sexes. Since tissue stiffness depends on both the cellular and ECM composition, as well as their structural arrangement [19,22,41,55,68,79], the combination of up- and downregulation of different matrix components may counterbalance their effect on mechanical properties and may even serve as a protective measure. Future studies with larger group sizes will allow for in-depth analyses of the individual contribution of matrix components to the viscoelastic behavior of the brain. Furthermore, as viscoelastic matrix properties have been shown to strongly influence neuronal and astrocytic growth as well as the differentiation of oligodendrocytes [80,81,82], it remains to be determined how the absolute sex differences in cortical stiffness affect cellular behavior and remyelination in EAE.

## 5. Conclusions

In conclusion, by applying multifrequency MRE we were able to demonstrate, for the first time, significant sex differences in viscoelastic properties of the cortex of healthy and EAE mice. Softening was not associated with lesion formation but with ECM remodeling, as reflected by the gene expression changes observed for collagen type IV, laminin, and fibronectin. Understanding sexual dimorphism in the composition of the ECM is important in neurobiology, and might help understand sex-dependent responses to environmental factors, as well as provide guidance for the development of individualized treatments of MS. Moreover, possible implications of sex-specific mechanical properties on the cellular behavior in neuroinflammatory conditions need to be investigated. Regarding future imaging studies, sex-specific reference values for MRE in MS should be established. 

## Figures and Tables

**Figure 1 biology-11-00230-f001:**
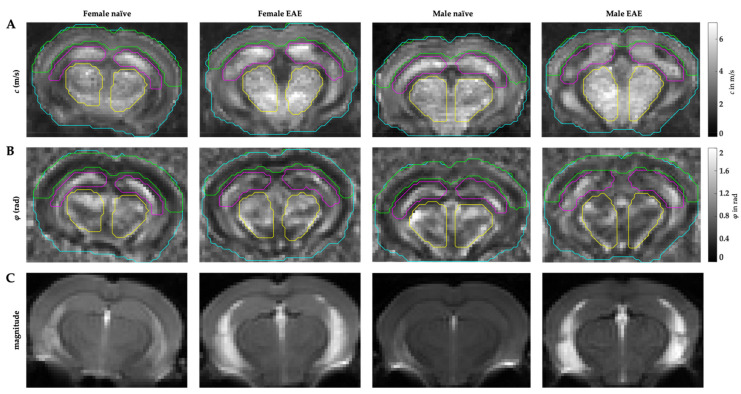
Representative parameter maps of the coronal murine midbrain with defined regions of interest in female/male naive and female/male EAE. (**A**) stiffness maps in *c* (m/s), (**B**) fluidity-maps in *φ* (rad), (**C**) anatomical magnitude images (arbitrary units). Cyan = whole midbrain, green = cerebral cortex, magenta = hippocampus, yellow = thalamic area.

**Figure 2 biology-11-00230-f002:**
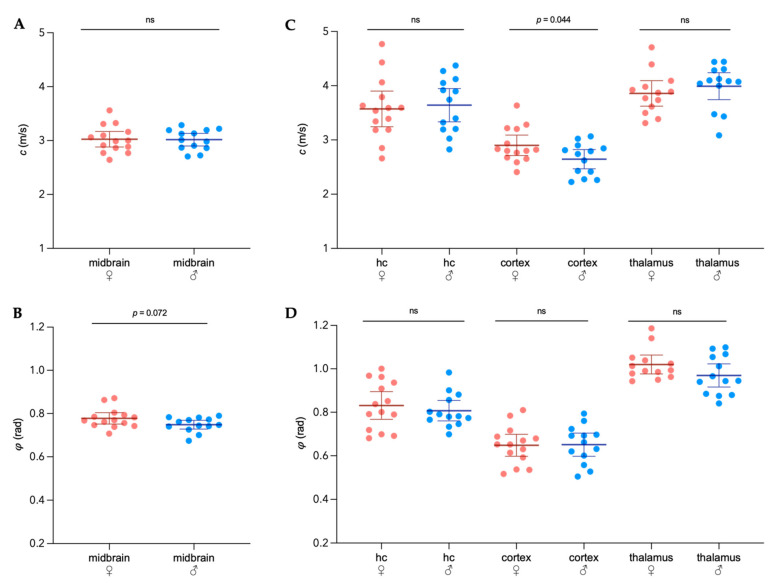
Sexual dimorphism in viscoelastic properties of healthy adult mouse brains. Mean shear wave speed *c* in m/s (**A**) and fluidity *φ* in rad (**B**) in the whole coronal section of the midbrain of female and male mice. (**C**) Regional analysis of stiffness (*c*) between sexes in the cortex, hippocampus, and thalamus. (**D**) Regional assessment of fluidity (*φ*). *n*_female_ = 14, *n*_male_ = 13, hc = hippocampus; representation of individual data points with mean and 95% CI. Group comparison performed by *t* test. ns = not significant.

**Figure 3 biology-11-00230-f003:**
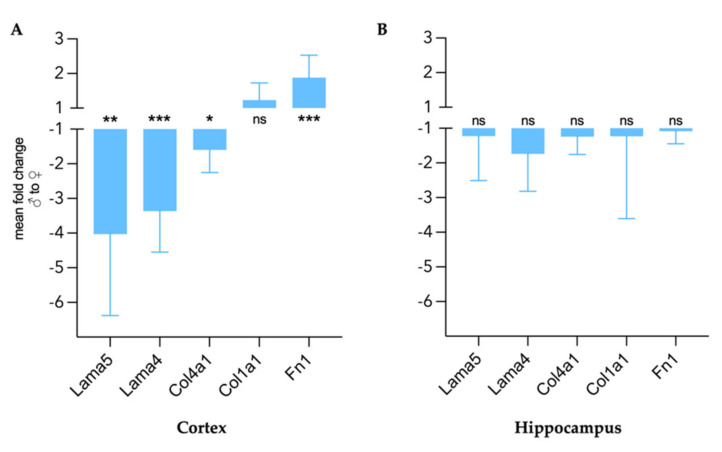
Sex differences in gene expression of ECM components in healthy cerebral cortex (**A**) and hippocampus (**B**) expressed as mean fold change of male gene expression compared to female with 95% CI. *n*_female_ = 6, *n*_male_ = 6. Group comparison performed with unpaired *t* test with Welch correction. *Lama5* = laminin *α*5, *Lama4* = laminin *α*4, *Col4a1* = collagen type IV *α*, *Col1a1* = collagen type I *α*1, *Fn1* = fibronectin 1. * *p* < 0.05, ** *p* < 0.01, *** *p* < 0.001, ns = not significant.

**Figure 4 biology-11-00230-f004:**
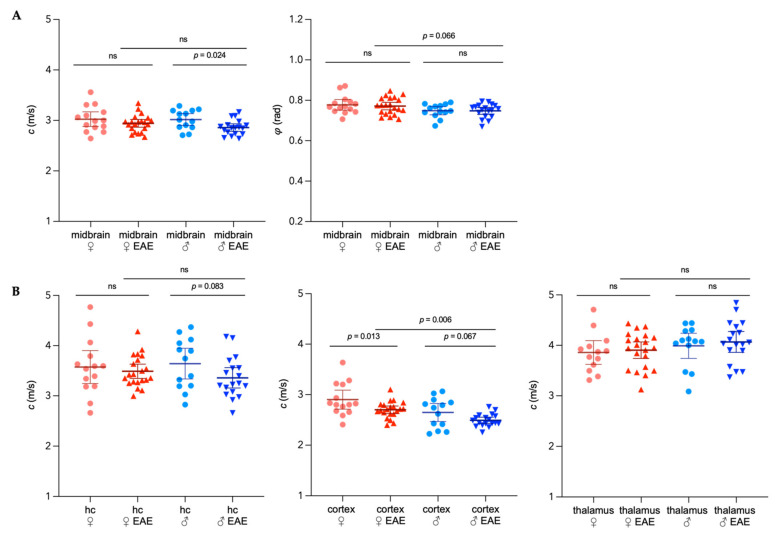
Sex-specific viscoelastic properties of the mouse brain in healthy conditions and EAE. (**A**) Shear wave speed *c* in m/s and fluidity *φ* in rad in the whole coronal midbrain. (**B**) Regional distribution of *c* in the hippocampus (hc), cortex and thalamic area; light red = healthy female; dark red = EAE female; light blue = healthy male; dark blue = EAE male. Representation of individual data points with mean and 95%CI. *n_female naive_* = 14, *n_male naive_* = 13, *n_female EAE_* = 21, *n_male EAE_* = 18. Two-way ANOVA with Fisher’s LSD post-hoc comparisons. ns = not significant.

**Figure 5 biology-11-00230-f005:**
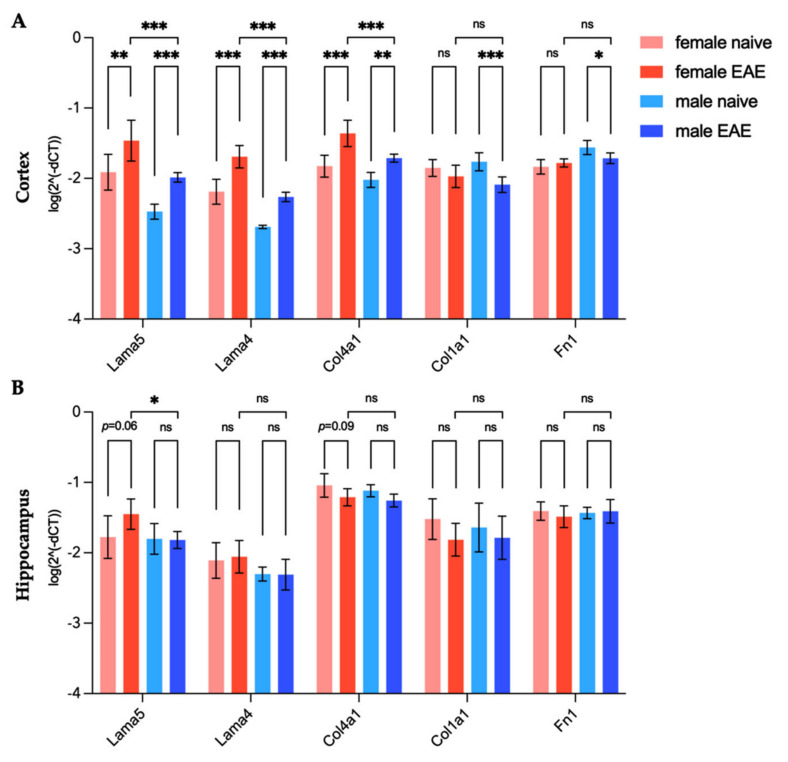
Sex-specific gene expression of ECM targets in healthy and EAE mice in the cortex (**A**) and hippocampus (**B**) relative to endogenous reference *Hprt1*. Representation of mean log^[2^(−ΔCT)]^ with 95% CI. *n*
_female naive_ = 6, *n*
_male naive_ = 6, *n*
_female EAE_ = 6, *n*
_male EAE_ = 7. Two-way ANOVA with Tukey’s post-hoc comparisons. * *p* < 0.05, ** *p* < 0.01, *** *p* < 0.001, ns = not significant.

**Figure 6 biology-11-00230-f006:**
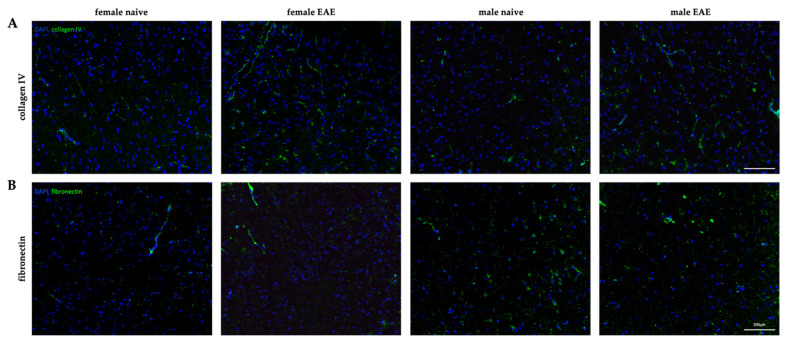
Sexual dimorphism in cortical ECM remodeling in EAE. (**A**) Immunofluorescence of collagen type IV (green) in healthy condition (naive) and at peak EAE for female (left) and male (right) cortices. (**B**) Cortical fibronectin (green) staining of female (left) and male (right) healthy and EAE mice. DAPI (blue). 20× magnification. Scale: 100 µm.

**Table 1 biology-11-00230-t001:** Sex-specific fold changes in gene expression of ECM components in EAE compared to healthy and between EAE in cortex and hippocampus.

	Cortex	Hippocampus
Gene	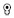 EAE/Naive	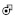 EAE/Naive	EAE 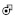 / 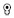	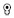 EAE/Naive	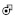 EAE/Naive	EAE 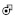 / 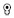
*Lama5*	2.91	3.04	−3.85	1.83	−1.08	−2.42
*Lama4*	3.10	2.70	−3.86	1.08	1.11	−1.69
*Col4a1*	2.99	2.01	−2.38	−1.51	−1.37	−1.12
*Col1a1*	−1.29	−2.12	−1.34	−2.07	−1.33	1.26
*Fn1*	1.12	−1.43	1.18	−1.19	1.14	1.24

## Data Availability

The raw data supporting the conclusions of this article are available upon request from the corresponding author.

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
