# Peer review of "Sexual Dimorphism in Extracellular Matrix Composition and Viscoelasticity of the Healthy and Inflamed Mouse Brain"

_biology, 2022, doi:10.3390/biology11020230_

Round 1
Reviewer 1 Report
Batzdorf and colleagues have written an original article on the sex-dependent extracellular matrix decomposition in healthy and inflamed brains. As the mechanobiology of the brain recently has attracted a lot of attention and the study provides a lot of mechanobiological and gene expression measurements, the topic and data are quite interesting and timely. However, there are a few points to be addressed:
- Regarding qPCR data shown in Figure 5: according to the design of experiment, the appropriate statistical test is two-way ANOVA followed by post-hoc pairwise comparison between groups.
- Moreover, having data for elasticity and ECM expression for multiple brain regions allows a regression analysis to be performed using a general linear model. It may reveal which of genes would allow to predict individual regional viscoelasticity measurements, by considering c as a linear function of gender, condition, and measured gene expression values (can be optimally selected using stepwise regression analysis). Possibly, interactions between some of these factors could be added. This analysis would allow to estimate which % of variability in viscoelasticity can be explained by the expression of the studied genes (R2), and select a combination of the most informative genes.
- Please cite studies characterizing the selected EAE model for the selected mouse strain, and briefly summarize its neuroinflammatory features, particularly in the cortex.
- The authors compared the qPCR results with staining for Fibronectin and Col4 but the representative images do not match the qPCR results. For example: In Figure 6, in the representative image of fibronectin staining in male naive and EAE, there is a huge downregulation of fibronectin while in qPCR data the difference is very mild. The qPCR data from figure 5 shows a highly significant difference (***) in Col4 expression in the male naive and male EAE group while the representative images are not reflecting this difference. How consistent was a pattern of fibronectin staining as shown for the male control mouse? (here are many stained cells in addition to vessels). Please make sure that really representative images are shown.
- Line 307: The conclusion about the role of sex-dependent basement membrane ECM in overall stiffness and inflammatory responses is far too simplified and too speculative.
- The authors mentioned laminin, collagen, and fibronectin as important regulators of the mechanical properties of the brain ECM. It is clear that these molecules are ''important regulators of the mechanical properties of the basement membrane ECM''. It is unclear, however, how constituents of this small fraction of brain ECM may regulate the rest of neural tissue, including perisomatic and perisynaptic neural ECM. More discussion is necessary to address this major point. One aspect is that the changes in the mentioned proteins may just correlate with changes in the state of endothelial cells, astrocytes, and OPCs and these cells synthesize major components of neural ECM, like lecticans and tenascin-R, hyaluronic acid synthases, metalloproteinases and their tissue inhibitors, which would indeed regulate the stiffness of neural ECM. Additionally, laminins, collagens and fibronectin may be incorporated into neural ECM and form the hybrid-like ECM (https://pubmed.ncbi.nlm.nih.gov/32594588/), which impact may be underestimated.
- Line 320, the authors mentioned ''Since these components have been shown to be mechanically relevant'' without referring to original papers highlighting this fact.
- Please briefly explain in the introduction why multifrequency magnetic resonance elastography (having a higher noise sensitivity) was used and how these measurements are related to brain stiffness measurement provided by other approaches like atomic force microscopy?
- Why does Hprt serve as the endogenous reference? Please give a reference.
- Please explain in one sentence what is shown as the magnitude in Fig. 1, which seems to be highly increased in EAE male and female in a specific area. Also, visually there is a strong upregulation of c in that area. It was not, however, analyzed there at all. Please explain why.
- Line 199-200, please always add units after numbers.
- Figure 3: please add SEMs to bars.
- Different numbers of EAE males and females were used for the study, is it by chance or some animals were excluded from the analysis? Then, the reasons and number of animals should be mentioned in the methods.
Reviewer 2 Report
In this paper, the authors showed significant sex differences in viscoelastic properties of the cortex of healthy and EAE mice. Softening was not associated with lesion formation but with ECM remodeling as reflected by the gene expression changes observed for collagen type IV, laminin, and fibronectin. Overall, this paper is well written with sufficient introduction, detailed methods and solid data. The topic is both interesting and important and will be helpful for future studies on the effects of sex differences on the brain. However, the discussion part is relatively weak. The authors found significant viscoelastic property changes in the cerebral cortex, but not in the thalamus. In fact, the thalamus is a large structure and can be divided into multiple subregions. Subtle structural changes may be evident if the authors take a close look at the brain regions, the pulvinar nucleus in particular. Previous studies have shown that pulvinar is mutually and extensively connected with the prefrontal cortex, sensory cortex, superior colliculus and amygdala and plays very important roles in contextual multi-sensory processing and emotional response (PubMed ID: 32142411, PubMed: 31812514). In addition, the changes of pulvinar has been reported to be associated with sexual dimorphism (PubMed ID: 34506914). The authors should include the above key citations in the discussion. I would like to recommend this interesting paper to the editor after revision.
